# Stress, Coping, and Adjustment of International Students during COVID-19: A Quantitative Study

**DOI:** 10.3390/bs14080663

**Published:** 2024-08-01

**Authors:** Ying Wu, Yi Ding, Tamique Ridgard, Akane Zusho, Xiaoyan Hu

**Affiliations:** Graduate School of Education, Fordham University, New York, NY 10023, USA; ywu135@fordham.edu (Y.W.); tridgard@fordham.edu (T.R.); zusho@fordham.edu (A.Z.); xhu118@fordham.edu (X.H.)

**Keywords:** COVID-19, pandemic, international students, university students, visa regulation, discrimination, stress, coping

## Abstract

The COVID-19 pandemic in 2020 caused unexpected and unprecedented disruptions worldwide. University students, especially international students, underwent unique challenges during this volatile era. This secondary quantitative data analysis study aimed to investigate the experiences, stress, adjustment, and ways of coping of university students during the COVID-19 pandemic. Data from 112 international university students and 112 domestic American university students were included in the statistical analyses. The differences in Student Adaptation to College Questionnaire (SACQ), Ways of Coping Questionnaire (WAYS), COVID-19-related questionnaire, and Perceived Stress Scale (PSS) between international and domestic students were analyzed using independent samples t-tests. Multiple regression models predicting PSS by SACQ, WAYS, and COVID-19-related questionnaire subscales were estimated for international and domestic students separately. The results showed that international students and American students did not report significant differences in their university adjustment, usage of coping strategy, and perceived stress level during the peak of the pandemic. Additionally, American students reported more factors impacting their perceived stress than international students. Implications of the findings as well as limitations and future directions were discussed.

## 1. Introduction

The COVID-19 pandemic has undoubtedly imposed disruptions to society [1,2]. According to past research, school closures were found to be influential in delaying the epidemic peak [3]. However, such disruption in learning inflicted elevated stress, anger, fear, and uncertainty on students [4,5]. Several studies conducted on university students’ remote instruction experiences discovered that students tended to report the online learning transition negatively, as the courses became less enjoyable, the lessons required less effort and attention, and the contents incorporated less cultural considerations [6,7].

Challenges with technology and connectivity were also repeatedly reported across studies, with students detailing their main struggle with connecting and maintaining internet quality to access lectures and materials [8,9]. Students who reported greater financial hardship since the beginning of the pandemic also reported significantly higher difficulty with internet connectivity, with obtaining and accessing technology devices, and with communication with faculty [10]. Some other difficulties included increased academic workload [8], suspended fieldwork, internships, and clinical rotations [11], canceled plans, internships, and employment, delayed or canceled graduation ceremonies [12], difficulty finding a quiet study space, and struggles with limited financial resources [7,9,10].

University students also reported high levels of distress due to academic difficulties and social isolation. Students were experiencing challenges with staying motivated and focused on academics, maintaining emotional well-being, feeling more exhausted and cynical, and decreasing self-efficacy [6,7,9]. Underrepresented minority students also held more negative attitudes towards the transition and experienced more difficulty in assignments [7]. Students reported higher anxiety and distress in general, decreased social engagement and connection, lack of motivation, and decreased maintenance in healthy habits [6,9]. Because of their already high level of distress, the psychological effect of social distancing may be intensified in university students [13]. 

As an integral part of university compositions, international university students studying in the United States on student visas unfortunately faced additional difficulties compared to their domestic counterparts. Such struggles included but were not limited to, visa complications in terms of suspended routine visa services and volatile and unpredictable visa guidelines [14,15,16]; disrupted travel and education plans due to exponentially rising air-travel fare, decreased flight availability, strict travel restriction guidelines between different countries, and disrupted fieldwork and research requirements [17,18,19,20]; time zone differences and technological difficulties in relation to remote learning [21,22,23]; housing and financial complications in terms of limited warning to closed dormitory, difficulty with searching and renting places to live within short time frames as foreign nationals without applicable income, and work and financial aid restrictions due to visa status [14,18,24,25]; and uncertainties to future and safety due to aforementioned struggles [14]. Although some support measures were temporarily put in place to help international university students navigate through their difficulties, these measures were not enough to account for the specific needs and situations that these students were forced to be in due to the nature of their legal status. 

As indicated in recent literature, the delay in universities’ action to address discriminatory incidents and changes in student visa policies further fueled international students’ feelings of mistrust, disappointment, and helplessness towards their institutions and even towards the American society in general [26,27]. The Student and Exchange Visitor Information System (SEVIS) by the Number Report published in 2020 and 2021 showed a decrease in the number of international students enrolled in American schools since 2019 [28,29], which may impact the international students that are currently enrolled in universities by reducing the already limited attention and resources placed on the welfare of this population. Therefore, it is imperative for higher education institutions to promptly investigate and address this potential mismatch existing in the experiences and needs of their international student communities, especially during an unprecedented time like the COVID-19 pandemic, as well as to prepare for future events such as natural disasters, international conflict, and so on. 

### 1.1. University Adjustment

As discussed, the extraordinary disruptions caused by the COVID-19 pandemic have drastically changed the perspectives of higher education [4]. Students are expected to overcome a series of complex challenges that require emotional, social, and academic adjustment demanded by the pandemic, the university, and their new ways of life [30]. Adjustment to university demands is closely associated with a successful overall college experience [31]. As defined by multiple studies, university success is best measured through the combination of the students’ cognitive capacity and academic achievement. The failure to adjust to university demands may lead to student attrition [30,32,33,34]. Baker and Siryk [34] identified four factors closely related to university adjustment, namely academic adjustment, social adjustment, personal-emotional adjustment, and institutional attachment. They further designed a quantitative measurement to assess how well a student is adapting to the demands of the college experience, which is known as the Student Adaptation to College Questionnaire [34].

Academic adjustment refers to students’ ability to adapt to the educational demands in their university experiences. It closely correlates with students’ purposeful planning of educational goals, self-appraisal, academic self-efficacy, and academic-related skills [30,34]. Personal-emotional adjustment focuses on students’ psychological and physical well-being during their adjustment, as well as their coping skills, perceived distress level, and emotional reliance on others [34]. It is impactful to students’ general school performance, overall functioning, and ability to cope with stress. 

Social adjustment refers to students’ ability to cope with interpersonal–societal demands inherent in their university adjustment experiences. Social support networks are extremely crucial to students’ university adjustment. Empirical research has emphasized the importance of students’ integration into the social environment since such integration is a crucial element in commitment to a particular academic institution. Students’ social adjustment, like academic factors, can be used to predict students’ persistence [30,34]. Students with better social adjustment tend to report a higher level of commitment to their academic goals and their institutions, fostering greater overall success in their university experiences [31,33,34].

Lastly, attachment refers to students’ degree of commitment to educational–institutional goals and the quality of relationship established between the students and their institution [34]. Research has found that university attachment is closely related to student attendance and feelings toward the institution, in which the greater the attachment and satisfaction to their university, the better the students’ perception of their social connection, acceptance, and academic competence [35,36].

International students, in particular, generally have been found to face a more complicated process of university adjustment. Research has found that many international students feel isolated from their university communities. Asian international students, in particular, were historically less likely to agree that their intelligence and hard work were acknowledged by their peers and advisors, leading to an intensified feeling of invisibility than students from other continents [37]. The inevitable perplexing acculturation process as well as frustrations in language proficiency and social interactions also placed limitations on international students’ academic and social adjustments [38,39,40]. Also, international students often have to adapt to a different educational system that requires different study methods and social skills. These stressors altogether can make the university transition experience even more stressful for international students [41]. Moreover, studies find that minority international students often feel like they are being treated as uninvited guests, and have experienced neo-racism or other forms of discrimination based on cultural differences and national order in their university environment. International students also report feelings of inferiority and experiencing lack of support to their transitions after enrollment, leading to a magnified dissatisfaction towards their institutions, communities, and college experiences [42].

Although the standardization of SACQ for college students was based on samples from universities across North America, consideration of race was primarily limited to Black and White students [34]. Research has explored the effect of race on student university adjustment and found a lack of difference in SACQ scores between racial and ethnic groups. Also, the different Asian identities were found to have no significant effect on overall college adjustment on SACQ scores [43,44,45], indicating a satisfactory usage of SACQ in measuring university adjustment for the present study.

### 1.2. Stress

Stress emerges when an individual’s perceived demands of the situation are beyond their own capacity to deal with the circumstances [46]. Perceived stress is determined by an individual’s feeling about the general stressfulness of their life and their ability to work through such stress during a given period of time [46,47]. In terms of the COVID-19 pandemic, although all university students were experiencing similar unprecedented stressors [13], each individual perceives the same stressors differently based on their perspectives and personal qualities. Research has found that perceived stress is negatively correlated with self-efficacy, with a higher degree of perceived stress indicating a lower level of self-efficacy [48,49]. Therefore, even if university students collectively experienced the impact of the COVID-19 pandemic, their thought process will determine the level of stress they perceived in this shared experience.

### 1.3. Ways of Coping

Coping is defined as the constantly changing cognitive and behavioral efforts that a person engages in to manage stressful external and/or internal demands [46,50]. The widely acknowledged transactional model of stress and coping, proposed by Lazarus and Folkman [50], explained the process of individuals’ usage of cognitive interpretations and behavioral responses to cope with the stress that they perceived from environmental events. Three major categories of coping styles were identified, including emotion-focused coping, problem-focused coping, and avoidance-focused coping. It is found that people who more often engage in proactive problem-focused coping and some positive emotion-focused coping generally experience less stress, while those who use more reactive emotional-focused coping experience more stress [51].

Research has found cultural differences between the usages of coping strategies, such that different cultural groups have different cognitive appraisals of stressful events, indicating that the prevailing cultural beliefs may impact the judgment of threat and availability of coping resources. For example, when facing stressful encounters, Japanese students were significantly more likely to use emotion-focused strategies such as escape-avoidance and positive reappraisal than British students [52]. Similarly, compared to European American youth, Chinese American youth tend to utilize problem-focused and avoidance-focused coping behaviors to moderate the effect of stress on negative adjustment [53].

### 1.4. Purpose of This Study

Through secondary analysis of an existing dataset, this study examined the perceived stress, ways of coping, and university adjustment of international university students studying in the United States during the COVID-19 pandemic. Although there is growing research on the effect of COVID-19 and university students’ well-being, limited research has examined the unique experiences of international students. This study aimed to explore the variety of experiences that international students encountered during the pandemic by comparing the experiences between international and domestic American university students. By incorporating the university adjustment, perceived stress, ways of coping, and COVID-19 factors, this study aims to answer the following research questions: (1) Do international and American university students report different levels of college adjustment and ways to cope with stress? (2) Given the extra struggles of acculturation, homesickness, and specific stressors introduced during the COVID-19 pandemic, do international students perceive higher levels of stress than American students under the same COVID-19-related situation? (3) Do college adjustment level, ways of coping, and the impact of the COVID-19 pandemic predict the perceived stress level differently between international and American students?

Accordingly, the hypotheses were as follow: (1) the mean SACQ and WAYS scores are different between international and domestic American university students, with American students reporting higher levels of adjustment and utilizing more coping strategies; (2) international students report a higher mean perceived stress scale score compared to their domestic counterparts; and (3) SACQ, WAYS, and COVID-19-related factors predict the level of perceived stress in international and domestic American university students differently, with international students reporting more factors impacting their stress level compared to their domestic American counterparts given their unique struggles.

## 2. Method

### 2.1. Participants

The original study included undergraduate and graduate domestic American and international students currently studying in the United States. Participants were eligible if they were at least 18 years of age, studying at an American university at either the undergraduate or graduate/professional level, and consented to participate in the study on a voluntary basis. Data collection occurred from March to early April 2020 via convenience sampling from professional networks of professors and students.

A subsample of the original large-scale study was extracted for the purpose of the current study. In the original study, compared to the larger participation of domestic American university students (*N* = 399), only 116 international university students participated. Therefore, all international students data were included for analysis in the current study to ensure a thorough representation. Upon analyses, 4 international students were excluded from the study due to incomplete survey responses, leaving a total of 112 international students included in the quantitative analyses. To match the number of the international student participants, a total of 112 American students were selected from the original study. They were selected to be proportional to the racial composition of college students in the United States [54] as well as the gender and age-group compositions of the international students. Altogether, the participants’ ages ranged from 18–51 years old (*M* = 25.04). Approximately 75% of the participants were female; around 64% were undergraduate students. See Table 1 and Table 2 for complete socio-demographic characteristics.

### 2.2. Measures

#### 2.2.1. Socio-Demographic Questionnaire

The original study included socio-demographic questions to gather information on participants’ age, gender, family income, race, student visa status, university locations, majors, degrees, and class year.

#### 2.2.2. Student Adaptation to College Questionnaire (SACQ)

The SACQ was included to investigate the processes and outcomes of students adapting to the demands of university experiences. This self-report questionnaire contains 67 items on a 9-point Likert scale with choices ranging from “1 = applies very closely to me” to “9 = doesn’t apply to me at all”. Participants’ responses were scored with appropriate items reversed coded, and then converted into T-scores for interpretation of the participant’s level of adjustment to university (30 = very low, 40 = low, 50 = average, 60 = high, 70 = very high). Four subscales were examined upon analysis. The social adjustment subscale contains 20 items (original large-scale study (OLS) α = 0.91) and measures students’ interpersonal–societal demands in their university adjustment experiences. The personal-emotional adjustment subscale contains 15 items (OLS α = 0.87) and measures students’ psychological and physical well-being. The attachment subscale contains 15 items (OLS α = 0.90) and measures students’ level of affection and sense of belonging to their universities as well as the relationship quality between the students and their institutions. The academic adjustment subscale contains 24 items (OLS α = 0.88) and measures students’ adaptation to the variety of educational demands in their university experiences [32].

#### 2.2.3. Perceived Stress Scale (PSS)

The PSS was used to examine the students’ perception of stress during the COVID-19 pandemic. It contains 10 items on a 5-point Likert scale with choices ranging from “0 = never” to “4 = very often” (OLS α = 0.87). The questions are content-free with focus on the participants’ thoughts and experiences in the present time of life, allowing the possibility to apply the questionnaire to practically any subpopulation. Participants are believed to be experiencing higher levels of stress if their accumulated scores are higher [47].

#### 2.2.4. Ways of Coping Questionnaire (WAYS)

The WAYS was included to explore the students’ coping strategies and process during the COVID-19 pandemic. It contains 66 self-report items on a 4-point Likert scale with responses ranging from “0 = does not apply or not used” to “3 = used a great deal”. Eight subscales were examined upon analysis [46], and each subscale derived a raw score. Participants’ scores were calculated for each subscale to indicate their coping strategies utilized during COVID-19, with a higher raw score indicating high frequency in usage of the coping behaviors in this category compared to other subscales.

The confrontive coping subscale (OLS α = 0.66) measured the participants’ aggressive efforts to change the situation to the point of being risky, while the distancing (OLS α = 0.64) subscale measured their efforts to remove themselves from the situation. The seeking social support subscale (OLS α = 0.67) measured the participants’ degree of attempt to reach out to their surroundings for consultation and comfort, whereas the accepting responsibility subscale (OLS α = 0.70) measured their efforts on reclaiming self-worth and engaging in positive changes in thoughts and behaviors. The positive reappraisal subscale (OLS α = 0.74) measured the efforts to create positive meanings by focusing on personal growth, and the planful problem solving subscale (OLS α = 0.71) measured the deliberate efforts to focus on the problem itself to change the situation. Lastly, the escape-avoidance subscale (OLS α = 0.67) measured participants’ wishful thinking and behavioral efforts to escape or avoid the situation, whereas the self-controlling subscale (OLS α = 0.64) measured participants’ efforts to regulate their own feelings and actions [50].

#### 2.2.5. COVID-19-Adjustment Questionnaire (COVID-19 AQ)

A 37-item, 5-point Likert scale self-report questionnaire, with choices ranging from “1 = strongly disagree” to “5 = strongly agree”, was created for the original study to measure the impact and adjustment of COVID-19 on participants. Responses were scored following the scoring procedure of the SACQ in terms of how to score for omitted questions and accumulate item scores for subscale results. This COVID-19 adjustment measure was adapted from an unpublished instrument created to measure the mental health index and experiences of university students during the initial outbreak in China [55]. For the original large-scale study, factor analyses on the questionnaire yielded five domains (Kaiser–Meyer–Olkin Measure of Sampling Adequacy = 0.81; χ^2^(325) = 787.90, *p* < 0.001), with one excluded due to low reliability (OLS *α* = 0.38).

The remaining four domains, including the adaptive adjustment subscale, the social support subscale, the academic adjustment subscale, and the discriminatory impact adjustment subscale, were calculated separately to measure the participants’ level of adjustment to the COVID-19 pandemic. The 6-item adaptive adjustment subscale (OLS *α* = 0.76) measured the participants’ ability to handle the impact and feelings related to their COVID-19 experiences. The 4-item social support subscale (OLS *α* = 0.69) measured the participants’ satisfaction level towards the social support they received during the pandemic. The 8-item academic adjustment subscale (OLS *α* = 0.85) measured the participants’ preparedness and motivation to adjust and complete academic requirements under new circumstances. Finally, the 3-item discriminatory impact adjustment subscale (OLS *α* = 0.78) measured the participants’ knowledge and acknowledgement of racist incidents that arose during the COVID-19 pandemic. Although all subscales on this COVID-19 adjustments questionnaire contained less than 10 items, they all demonstrated Cronbach’s alpha levels above 0.50, which were considered as sufficient to use for statistical analysis [56]. Participants are indicated to be adjusting more positively during the COVID-19 pandemic if they received a high score on these subscales.

### 2.3. Procedures

An Institutional Review Board (IRB) approval was obtained from Fordham University prior to original data collection. Participants were recruited through convenience sampling from professional networks of professors and students. Survey invitation and the link to the Qualtrics survey were sent to interested university students. An electronic consent form was presented at the beginning of the survey for all participants to read and sign voluntarily before they could access the survey materials. The titles and labels of all self-report measures were removed for the participants, and each measure was presented and completed in full before the participants could access the next measure. Eligible and interested participants provided consent and completed the self-report survey with an average of 25-min completion time. Only participants who completed the survey entirely were included for data analysis of the original study. Numerical IDs (i.e., 001, 002) were assigned to each participant and all identifiable information was removed.

For this current secondary data analysis study, an IRB approval from Fordham University was obtained prior to data analysis. This current study extracted subsamples from the original study population, specifically 112 domestic American university students and 112 international university students, for the purposes of comparison.

## 3. Results

### 3.1. Descriptive Analyses of Variables

Descriptive analyses were conducted to describe the participant characteristics of the subsamples included in this current study (see Table 1 and Table 2).

Descriptive statistics for SACQ, WAYS, PSS, and COVID-19 AQ are presented in Table 3. For international university students, there were skewness found in the following variables: SACQ social adjustment, WAYS confrontive coping, and COVID-19 adaptive adjustment, discriminatory impact adjustment, and the international student specific question. For American students, skewness was found in variables SACQ personal-emotional adjustment, WAYS confrontive coping and accepting responsibility, and COVID-19 adaptive adjustment and discriminatory impact adjustment.

### 3.2. Data Analysis

The internal consistency of each subscale of all questionnaires measured was examined through the calculation of Cronbach’s alpha based on the results yielded from the original large-scale study, in which all were deemed acceptable. Statistical assumptions were checked by analyses of normal distribution, skewness, and equality of variances, which yielded no violations.

The first research question explored whether there were differences between international and American students in their adjustment to college and different ways to cope with stress. The second research question asked if international students would perceive higher levels of stress than American students under the same COVID-19-related situation given their unique additional struggles. To address both questions, a multivariate analysis of variance (MANOVA) test was conducted to investigate whether or not there were mean differences in SACQ, WAYS, and PSS scores between international and domestic American students. The Box’s Test of Equality of Covariance Matrices was examined, in which the assumption of homogeneity of covariance was not violated (*p* = 0.62), indicating that the observed covariance between the factors was equal. The Pillai’s Trace test indicated a significant difference in SACQ, WAYS, and PSS mean scores between international and domestic American university students, F(14, 209) = 0.229, *p* < 0.001.

The MANOVA results (see Table 4) indicated significantly higher mean scores in American students’ SACQ academic adjustment, social adjustment and attachment than their international counterparts. International students reported significantly higher mean scores on WAYS confrontive coping, self-controlling, seeking social support, accepting responsibility, planful problem solving, and positive reappraisal than their American counterparts.

To further understand and identify the variables that best distinguish among the two student groups, a follow-up discriminant function analysis was conducted. Table 5 presents a summary of the univariate analysis. It was revealed that SACQ personal-emotional adjustment, WAYS distancing and escape avoidance, as well as perceived stress were the only four variables that did not produce significant differences between the student groups, which aligned with the results of the MANOVA. Multivariate analysis revealed a significant difference between the two student groups, λ = 0.747, χ^2^(13) = 62.76, *p* < 0.001, with an R^2^-canoncial = 0.253 (see Table 5). The predictive accuracy of the model was 71.9% and the cross-validation sample was 67.0%.

The third research question inquired the differences in the influence of college adjustment, ways of coping to stress, and the impact of the COVID-19 pandemic on the level of stress perceived by international students versus American students. To answer this question, two multiple regression analyses were conducted to explore the extent to which SACQ factors, WAYS factors, and COVID-19-related factors influenced the level of perceived stress in international and American students during the COVID-19 pandemic. Perceived stress score was entered as the dependent variable and SACQ factors, WAYS factors, and COVID-19-related factors were entered as the independent variables. Multicollinearity demonstrated acceptable variance inflation factor (VIF), indicating no correlations between independent variables. Results indicated that the predictors explained 51% of the variance for international students, and 64% of the variance for American students.

For international students, it was found that SACQ academic adjustment and personal-emotional adjustment, and WAYS escape avoidance, significantly predicted perceived stress. For American students, it was found that SACQ personal-emotional adjustment, and COVID-19 adaptive adjustment and academic adjustment, significantly predicted perceived stress. For both student groups, perceived stress was significantly positively correlated with WAYS escape avoidance, and negatively correlated with all SACQ factors, WAYS planful problem solving, COVID-19 adaptive adjustment, social support, and academic adjustment. Additionally, for American students, perceived stress was found to be significantly positively correlated with WAYS confrontive coping, seeking social support, and accepting responsibility, and negatively correlated with WAYS distancing (see Table 6).

## 4. Discussion

This study explored the relationship between international and domestic American university students’ adjustment to university, ways of coping, and experiences during the COVID-19 pandemic and perceived stress in the United States. The purpose was to understand whether there existed a difference in perceived stress due to the effect of student status, adjustment to university, ways of coping, and COVID-19 experiences.

The first hypothesis stated that international students would report higher mean scores on the SACQ and WAYS than American students, which was only partially supported. International students reported more difficulties experienced in academic adjustment, social adjustment, and attachment to the college environment than American students on the SACQ. This finding aligned with previous literature, in which international students were believed to be experiencing more problems in their adjustment to the American university’s educational demands, as well as fitting into the university community due to acculturation, language barriers, and cultural backgrounds. Numerous studies have shown that international students tend to experience a more stressful academic adjustment process in college due to challenges such as language difficulties, acculturation process, different lecture styles, new pedagogy, feelings of invisibility and inferiority, and discrimination [38,40,42,57]. This repeated finding indicates a strong need for universities to address international students’ difficulties with fitting into the community to help these students better adjust to their daily living and intensive studying in a completely foreign environment.

American students, on the other hand, reported higher use of the following coping strategies: confrontive coping, accepting responsibility, escape avoidance, and positive reappraisal than international students on the WAYS. The discriminant analysis result further revealed that the coping strategies of confrontive coping, accepting responsibility, and positive reappraisal are the top three variables that distinguished these two student groups. Given the large number of Asian students in the international student sample, this result did not align with past literature where students from Asian countries were more likely to engage in avoidance-focused coping strategies [41,52]. However, the previous literature provided a mixed suggestion regarding international students’ coping strategies, with some studies discovered that international students tend to practice problem-solving and seek for social support [57,58], whereas others may engage in avoidance-focused coping strategies [41,59]. Additionally, a recent study discovered that international students in the researchers’ targeted universities were exceptionally more vulnerable to experiencing greater sense of loneliness and poor mental health during the COVID-19 restrictions, with the feeling of being in a foreign environment exacerbating such experience. As such, these international students reported strengthening the connection with their families as an effective coping strategy [60]. The finding in this study, along with the mixed results from previous literature, underscored the importance of conducting thorough investigations on the coping styles exercised by the international student population to better account for cultural differences in coping strategies between students from different countries and institutions.

Although the second hypothesis stated that international students would report a higher perceived stress score than American students, the result showed that personal-emotional adjustment to university was the only factor that showed a significant difference between the two samples. Additionally, it was suggested that international and American students perceive the overall stressor of COVID-19 similarly and felt comparable about their own ability in handling the stressors in life during COVID-19. The significant difference between international and American students’ reported personal-emotional adjustment could be stemming from the diverse stressors experienced in their college adjustment. Previous literature have noted that international students, being one of the minority groups within university populations, experience complex layers of stressors when adhering to their universities’ demands. They are noted to have less access to resources such as the health care system, and are more prone to mental health disorders such as depression even under regular circumstances [61]. When coupled with the feelings of invisibility and inferiority, experiences of discrimination and misunderstanding, and the usage of ineffective coping strategies, it is possible that the international students may experience more intensive distress [40,41,57]. Nonetheless, this finding, in general, can be explained by the shared experiences of university students in their college environment. Regardless of their student status and cultural background, university students generally undergo similar experiences through their university and living environments, academic demands, extracurricular activities, and so on.

The third hypothesis proposed that international students would report more factors that impact their stress level than American students. However, the results showed that American students reported more factors impacting their stress levels. For both student groups, the findings suggested that university students may perceive lower levels of stress during the pandemic if they were better adjusted to their universities overall; if they focused on planning and solving the problems; and if they received more social support and were better adapted to the COVID-19 academically and in general. Moreover, the findings showed that the more engagement in escaping and avoiding the stressors, the more likely that university students would perceive higher levels of stress during the pandemic. These findings have been well discussed in the literature, in which university students who demonstrated better university adjustment generally experience lower levels of stress and better overall success [31]. Likewise, those who practice active and positive coping tend to report lower levels of stress than those who engage in negative and avoidance coping [62,63,64]. With this result in mind, higher education institutions are strongly encouraged to foster a welcoming environment and sense of community to their students to provide resilience factors against perceived stress levels. Universities should also partner with mental health professionals and communities to develop and implement systemic interventions and individualized supports that introduce college students to positive and proactive coping strategies.

Additionally, the results found that personal-emotional adjustment to college is a negative predictor of perceived stress during the COVID-19 pandemic for both international and American students. Supported by past literature, given that personal-emotional adjustment focuses on students’ psychological states and physical well-being during their transition to college, it influences students’ general academic performance, overall functioning, and ability to cope with stressors [30]. Therefore, it is likely that those who experience better personal-emotional adjustment to college, regardless of student status, may generally perceive lower levels of stress due to their well-rounded wellbeing.

Interestingly, only two additional predictors that impact perceived stress during the pandemic were found among international students. Specifically, academic adjustment to college is found to be a negative predictor to stress, while escape avoidance coping is a positive predictor. Research has indicated that Asian international students, in particular, were more likely to experience frustrations in problems with the university academic demands, and were likely to engage in escape-avoidance coping when encountering stressors [33,41,52,53], leading to aggravated stress levels [64]. Conversely, American students reported more predictors to their perceived stress. The finding suggested that the more usage of confrontive coping, seeking social support, and accepting responsibility, or the more distanced the students placed themselves away from the pandemic, the more likely that they would perceive more stress, which is again supported by the literature [62,64]. American students’ perceived stress is also negatively predicted by their adaptive and academic adjustment to the COVID-19 pandemic.

These differences in correlations and predictors between international and American students were quite interesting. It is possible that American students reported more correlations between coping strategies and stress level than their international counterparts because of the Western context of WAYS [65] and the cultural differences in coping [66]. Since the WAYS questionnaire was developed and standardized with the demographics in the United States, it is possible that it may not fully capture the specific coping behaviors practiced by international students due to cultural differences. The connections between increased levels of stress and adaptive and academic adjustment during the pandemic was also unique to the American students. Given the majority of the international students included in this study were from Asian countries, it is possible that they were already made aware of the impact of the COVID-19 pandemic from their family members back home, and hence they were able to adjust quicker and better when the outbreak first peaked in the United States in March 2020 [67]. The constant hardship that international students experience during their college and cultural adaptation, as well as the emphasis that they place on academic success due to their Asian backgrounds, may have also helped with their academic adjustment in general during the pandemic [68,69].

## 5. Limitations and Future Direction

Participants in this study were pulled from a larger sample pool, with all international students included and domestic American students strategically selected based on the information from the American Council on Education’s reported race [70]. However, it is important to note that the international students sample lacked a representation of Black students due to no report of the race among survey respondents. Additionally, the original study recruited participants through convenience sampling, which may lead to sampling bias and reduce the generalizability of the findings to the broader population. Future studies should aim to recruit participants more strategically and systematically to ensure an equal representation of all races and ethnicities among different student populations.

The survey data of 112 American students was strategically selected to represent the current race composition within college students studying in the United States, with only 11 students of Asian heritage being included. However, the number of Asians included in the international student sample was 94 out of 112. Given that the majority of international students studying in the United States are of Asian heritage [29], coupled with the discriminations and hardship that Asians and Asian Americans experienced due to the COVID-19 pandemic, it is difficult to separate the effect of student status and anti-Asian experiences on college students’ perceived stress during the COVID-19 pandemic. Future studies should consider matching racial composition (e.g., the number of students of Asian heritage) between international and American student samples to fully capture international students’ unique experiences and struggles that stemmed from student status. In addition, future research should account for additional sociodemographic variables within the international student sample, such as nationality, household income, gender, and years studying in the United States, when examining international students’ experiences during unprecedented global events (e.g., the COVID-19 pandemic) to understand whether these additional variables may explain international students’ experiences differently.

Besides referencing the race composition within college students studying in the United States, the American student sample was also selected strategically to match the gender and age of the international student sample. Other sociodemographic factors such as household income, school major, university level and years in school, and living arrangements were not considered, which may again impact the representativeness of the results. Future studies should consider matching sociodemographic profiles between American and international students to limit potential biases in the results as well as to examine how these additional sociodemographic factors may influence college students’ experiences and struggles during the COVID-19 pandemic and other global events. Future studies should also account for sociodemographic factors when examining their hypotheses to control for potential effects from these factors on the outcome.

The WAYS was noted to be one of the most widely respected and used coping questionnaires in the field of psychology for the last three decades [71,72,73]. However, recent literature has argued that the WAYS has rarely been applied during the current COVID-19 pandemic, and hence remains unclear whether it is a valid and comprehensive measure of coping behaviors relevant to an unprecedented, specific, global stressor [74]. Future studies are encouraged to investigate the validity and reliability of the WAYS in measuring the coping behaviors and effort specifically during the COVID-19 pandemic. Future studies should also consider developing or using a measure that is adaptive and capable of capturing the full range of coping behaviors during an unprecedented global event in the context of a pandemic.

In contrast, the COVID-19-related survey was created in relation to the emergence of COVID-19 and its global impact on daily lives within a short period of time. Despite conducting factor analyses to improve the survey quality, it did not thoroughly measure the four domains included in the analyses. More specific questions could be added to help explore additional aspects of university students’ COVID-19 experiences, and its consistency and validity need to be further investigated. More questions should also be included to specifically investigate international university students’ COVID-19-related struggles and experiences. Likewise, although the SACQ has been proven to be valid in measuring different aspects of university adjustments, there is a lack of data on diverse populations, such as international students, to form standardizations and explore cultural effects in university adjustments. The SACQ was originally standardized on predominantly White first-year undergraduate students in the United States [34]. Therefore, future research is highly encouraged to explore the validity and reliability of the SACQ on different cultural populations and students from different years of undergraduate and graduate programs [34].

As additional future research directions, studies are highly encouraged to develop specific institutional support for international students to improve their adjustment. Future studies can also further explore the differences in coping between international students from different countries in general and when placed under similar situations like the COVID-19 pandemic. This can help to clarify the nature of such differences (e.g., cultural and ethnic differences) and could help institutions to develop more comprehensive and sensitive support services. Future study can also explore the nature behind the shared experiences and common usage of coping strategies in university students’ encounter of situational and environmental stressors, regardless of student status and sociodemographic backgrounds, to understand the experiences shared within university students and to develop plans to better support students’ mental health.

## 6. Conclusions and Implications

This study investigated the university adjustment, perceived stress, and ways of coping of international and domestic university students studying in the United States during the COVID-19 pandemic. The results highlighted the differences in stressors, adjustments, and coping strategies between international and domestic American university students. Future studies are encouraged to continue the exploration of international students’ specific experiences in higher education in general, as well as under phenomenological events like the COVID-19 pandemic. This research is an important contribution to the psychological literature, as it can inform the development and implementation of support specifically for international university students in response to the COVID-19 pandemic and potentially to future unprecedented situations.

This study identified several coping strategies that international students utilized during the COVID-19 pandemic. Specifically, the results highlighted the prevalence of seeking social support in mitigating participants’ perceived stress, and adjustment to their universities and during the pandemic. Grant-Vallone et al. [33] found that university students who reported more peer support and greater usage of student support services indicated higher social adjustment, and those who reported better adjusted to campus life were more likely to feel attached and committed to their university. It is not uncommon to see international students moving away from seeking mental health support due to fear of stigmatization, preference to speak with those from the same background, lack of awareness to the existing mental health support, missing information received from their universities, and societal occupational or structural differences between their home country and the United States [67,75]. Given the complex layers of struggles in international students’ acculturation and university adjustment, it is imperative for universities to develop adequate and tailored supports to better address international students’ specific needs during their transitions and adjustment to college, new cultural environment, and to unprecedented events such as the COVID-19 pandemic.

Additionally, the results of this study highlighted several differences between previous research regarding coping strategies used by international student populations [38,41,52,53]. Given the rapid evolution in globalization in relation to economic developments and technological advancement, it is possible that the younger generations worldwide are being exposed to new and diversified values, including cultural values, religious beliefs, gender identifications, historical understanding, and more. This massive exposure to new content, coupled with the rapidly changing nature of the world, may expose individuals to new stressors and influence their beliefs and understanding of coping strategies. Therefore, it is imperative for research to keep pace with globalization and update its understanding of stressors and coping commonly seen in different populations, such as international students studying in the United States.

## Figures and Tables

**Table 1 behavsci-14-00663-t001:** Participant Demographic: Race (*N* = 244).

Race	*n*	%
Student Status		
International Students	112	50.0
White	12	10.7
Black or African American	0	0.0
Hispanic or Latino	2	1.8
Asian	94	83.9
American Indian or Alaska Native	0	0.0
Native Hawaiian or other Pacific Islander	1	0.9
Other	3	2.7
American Students	112	50.0
White	63	56.3
Black or African American	12	10.7
Hispanic or Latino	22	19.6
Asian	11	9.8
American Indian or Alaska Native	1	0.9
Native Hawaiian or other Pacific Islander	1	0.9
Other (e.g., Biracial)	2	1.8

**Table 2 behavsci-14-00663-t002:** Participant Demographic: Gender, Age, Household Income, University Level, University Location, Year in School, Major (*N* = 244; *n* = 112 Domestic American; *n* = 112 International).

	American		International		Total	
	*n*	%	*n*	%	*n*	%
Gender						
Male	26	23.20	29	25.90	55	24.55
Female	86	76.80	83	74.10	169	75.45
Age (Years)						
18–25	73	65.20	65	58.00	138	61.60
26–51	39	34.80	47	42.00	86	38.40
Household Income						
Less than USD 20,000	6	5.40	21	18.80	27	12.10
USD 20,000 to 49,999	33	29.50	40	35.70	73	32.60
USD 50,000 to 99,999	33	29.50	22	19.60	55	24.60
USD 100,000 and more	40	35.70	29	25.90	69	30.80
University Level						
Undergraduate	44	39.30	37	33.00	81	36.20
Graduate/Professional	68	60.70	75	67.00	143	63.80
University Location						
Metropolitan NYC area	50	44.60	39	34.80	89	39.70
Outside of Metropolitan NYC area	62	55.40	73	65.20	135	60.30
Year in School						
1–2	58	51.80	59	52.70	117	52.20
3–4	41	36.60	42	37.50	83	37.10
5–6 or higher	13	11.60	11	9.80	24	10.70
School Major						
STEM	16	14.30	35	31.30	51	22.80
Humanities	1	0.90	2	1.80	3	1.30
Social science	71	63.40	62	55.40	133	59.40
Medical or related field	14	12.50	/	/	14	6.30
Law	1	0.90	/	/	1	0.40
Business	2	1.80	7	6.30	9	4.00
Other	7	6.30	6	5.40	13	5.80

**Table 3 behavsci-14-00663-t003:** Descriptive Statistics for SACQ, WAYS, PSS, and COVID-19 AQ (*N* = 224).

	American (*n* = 112)				International (*n* = 112)			
Variable	Min	Max	*M*	*SD*	Min	Max	*M*	*SD*
SACQ								
Academic Adjustment	30	67	50.05	9.20	29	70	47.22	8.58
Social Adjustment	29	65	45.96	8.77	27	64	43.44	7.77
Personal-Emotional Adjustment	25	67	41.10	10.44	25	73	42.76	9.42
Attachment	32	73	48.96	8.47	32	60	45.81	7.22
WAYS								
Confrontive Coping	0	13	4.59	2.85	1	18	6.64	3.22
Distancing	1	16	6.71	3.22	1	18	7.37	3.22
Self-controlling	0	17	8.06	3.54	0	21	9.62	3.56
Seeking Social Support	0	17	7.43	3.22	0	18	8.85	3.51
Accepting Responsibility	0	10	3.40	2.55	0	12	4.90	2.71
Emotional Avoidance	2	20	10.34	3.98	1	24	9.89	4.37
Planful Problem Solving	0	15	7.03	3.31	0	18	8.71	3.30
Positive Reappraisal	0	16	6.26	3.87	1	21	8.26	3.82
Perceived Stress	5	33	20.52	6.51	6	33	19.22	5.51
COVID-19								
Adaptive Adjustment	6	28	12.86	0.45	6	30	14.32	4.29
Social Support	6	20	16.05	2.85	6	20	15.81	0.26
Academic Adjustment	7	35	16.80	6.23	7	34	17.95	5.73
Discriminatory Impact Adjustment	3	14	5.51	2.55	3	12	5.27	2.30

**Table 4 behavsci-14-00663-t004:** Multivariate Analysis of Variance of SACQ, WAYS, and Perceived Stress on Student Status.

	df	Mean Square	F	*Sig*.	Partial Eta Squared
SACQ					
Academic Adjustment	1	448.61	5.67	0.02 *	0.025
Social Adjustment	1	385.88	5.62	0.02 *	0.025
Personal-Emotional Adjustment	1	154.45	1.56	0.21	0.007
Attachment	1	556.29	8.98	0.00 **	0.039
WAYS					
Confrontive Coping	1	236.16	25.51	<0.001 ***	0.103
Distancing	1	24.45	2.36	0.13	0.011
Self-Controlling	1	136.72	10.84	0.00 ***	0.047
Seeking Social Support	1	112.86	9.96	0.00 **	0.043
Accepting Responsibility	1	126.00	18.18	<0.001 ***	0.076
Escape Avoidance	1	11.16	0.64	0.43	0.003
Planful Problem Solving	1	159.47	14.61	<0.001 ***	0.062
Positive Reappraisal	1	224.00	15.13	<0.001 ***	0.064
Perceived Stress	1	93.86	2.58	0.11	0.011

Note. * *p* < 0.05, ** *p* < 0.01, *** *p* ≤ 0.001.

**Table 5 behavsci-14-00663-t005:** Summary of Interpretive Measures of Discriminant Analysis.

Independent Variable	Unstandardized	Standardized	Structure Matrix (Rank)	Univariate F Ratio
SACQ				
Academic Adjustment	−0.071	−0.636	−0.258 (8)	5.671 *
Social Adjustment	−0.009	−0.073	−0.257 (9)	5.623 *
Personal-Emotional Adjustment	0.043	0.0432	0.135 (12)	1.562
Attachment	−0.032	−0.250	−0.325 (7)	8.984 *
WAYS				
Confrontive Coping	0.087	0.265	0.547 (1)	25.514 ***
Distancing	−0.047	−0.150	0.167 (11)	2.362
Self-Controlling	0.022	0.080	0.357 (5)	10.844 **
Seeking Social Support	0.087	0.292	0.342 (6)	9.959 **
Accepting Responsibility	0.140	0.368	0.462 (2)	18.177 ***
Escape Avoidance	−0.075	−0.312	−0.087 (13)	0.639
Planful Problem Solving	0.036	0.118	0.414 (4)	14.606 ***
Positive Reappraisal	0.026	0.100	0.421 (3)	15.129 ***
Perceived Stress	−0.012	−0.074	−0.174 (10)	2.582
Group Centroid International			0.579	
Group Centroid American			−0.579	
Wilks’ Lambda			0.747 ***	
(Canonical correlation)^2^			0.253	

* *p* < 0.05, ** *p* < 0.01, *** *p* ≤ 0.001.

**Table 6 behavsci-14-00663-t006:** The level of perceived stress predicted by SACQ, WAYS, and COVID-19 factors.

	International (*N* = 112)				American (*N* = 112)			
	*r*	*β*	*t*	*Sig*.	*r*	*β*	*t*	*Sig*.
SACQ								
Academic Adjustment	−0.53 ***	−0.23	−2.75	0.01 **	−0.43 ***	0.04	0.57	0.57
Social Adjustment	−0.28 ***	−0.04	−0.52	0.61	−0.22 **	−0.02	−0.23	0.82
Personal-Emotional Adjustment	−0.57 ***	−0.14	−2.34	0.02 *	−0.60 ***	−0.18	−3.31	0.00 ***
Attachment	−0.28 ***	0.17	1.59	0.12	−0.29 ***	−0.08	−0.76	0.45
WAYS								
Confrontive Coping	0.09	0.08	0.47	0.64	0.17 *	−0.02	−0.08	0.93
Distancing	0.01	−0.14	−0.81	0.42	−0.18 *	−0.14	−0.90	0.37
Self-Controlling	0.00	−0.19	−1.18	0.24	−0.08	−0.29	−1.87	0.06
Seeking Social Support	0.05	0.22	1.48	0.14	0.20 *	0.34	2.03	0.05
Accepting Responsibility	0.18 *	0.09	0.40	0.69	0.24 **	0.33	1.52	0.13
Escape Avoidance	0.43 ***	0.39	2.74	0.01 **	0.41 ***	0.22	1.67	0.10
Planful Problem Solving	−0.29 ***	−0.14	−0.70	0.49	−0.22 **	−0.27	−1.47	0.14
Positive Reappraisal	−0.06	−0.27	−1.68	0.10	0.19	0.01	0.04	0.97
COVID-19								
Adaptive Adjustment	−0.26 ***	0.08	0.66	0.51	−0.53 ***	−0.33	−3.02	0.00 **
Social Support	−0.25 **	−0.28	−1.56	0.12	−0.16 *	0.09	0.53	0.60
Academic Adjustment	−0.37 ***	−0.06	−0.64	0.52	−0.55 ***	−0.32	−3.55	0.00 ***
Discriminatory Impact Adjustment	−0.04	−0.26	−1.17	0.25	−0.13	0.06	0.32	0.75

* *p* < 0.05, ** *p* ≤ 0.01, *** *p* ≤ 0.001.

## Data Availability

Due to privacy concerns mentioned in the IRB protocol, the data associated with this study cannot be provided to the public without the supervision of the researchers. However, individual researchers who are interested in obtaining access to the data for individual use are encouraged to contact the corresponding author.

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
