# Peer review of "Stress, Coping, and Adjustment of International Students during COVID-19: A Quantitative Study"

_behavsci, 2024, doi:10.3390/bs14080663_

Round 1

Reviewer 1 Report

Comments and Suggestions for Authors

The onset of the COVID-19 pandemic in 2020 precipitated unforeseen disruptions worldwide, particularly impacting university students, including a notable cohort of international students facing unique challenges. This quantitative secondary data analysis study aims to explore the experiences, stress levels, adjustment processes, and coping strategies of university students amid the pandemic. By integrating data from 112 international and 112 domestic American university students, the study examines the differential effects of factors such as student visa status, adaptation to college, and coping mechanisms on perceived stress levels and adaptation to the pandemic. The findings offer valuable insights into the distinct experiences of international and domestic students, contributing to our understanding of the pandemic's implications for student well-being and academic success.

The authors are encouraged to revise certain aspects of the manuscript for clarity and accuracy:

In the introduction, it would be beneficial to clearly delineate the existing gap or deficiency in the current understanding or literature. By explicitly highlighting the lack of a particular situation or context, readers can better grasp the significance and relevance of the study's objectives and contributions. This clarity will enhance the overall comprehension and impact of the research presented.

While the concepts introduced in the introduction are solid, there is a need for stronger contextualization and relevance to both the broader scientific community and the readership. It would be beneficial to establish a clearer connection between the outlined concepts and their implications for current research trends or practical applications.

It's essential to explicitly state the type of study conducted in the methodology section for clarity and transparency. Case study approach? Including this information will help readers better understand the methodological framework used to investigate the research questions and interpret the findings within the context of individual cases

While the discussion provides valuable insights into the implications of the study findings, there is room for further elaboration and interpretation. It would be beneficial to delve deeper into the implications of the results and their significance within the broader context of the research topic. Additionally, consider reducing the descriptive elements of the results and instead focus on synthesizing the key findings to provide a more cohesive and comprehensive analysis. By engaging in a more detailed discussion, the authors can enhance the understanding of the implications of their research and contribute to the advancement of knowledge in the field.

Reviewer 2 Report

Comments and Suggestions for Authors

Thank you for the opportunity to provide a review of this article. The paper is well structured, the results support the conclusions, and the content references previous theoretical background.

The paper has several shortcomings that must be addressed before being considered for publication. 

 1.       2.1. Participants Section: Table 2 needs to present data separately for international students versus American students. It is important to see if they have similar sociodemographic profiles. For comparison, you should have selected 112 American students with a sociodemographic profile similar in terms of age, gender, and other relevant variables. Otherwise, the results could differ due to differing sociodemographic profiles. Additionally, Table 2 needs to be rearranged to reduce the line spacing, which is currently too large.

2.       In sections 2.2.2 and 2.2.3, why did you provide two values for the Cronbach's Alpha coefficient for each scale?

3.       For all scales that measure your concepts, please specify the minimum and maximum possible scores and explain the meaning of those scores.

4.       It is confusing in section 2.3, Procedure. You mentioned that the subsample contains 116 students, but in section 2.1, Participants, you mentioned a different number, 112.

5.       3.2. Section: Why did you name it "Quality Analysis" when it is a quantitative analysis?

6.       In section 3.2, line 321, you wrote, "results (see Table 5) indicated significantly higher mean scores in American students’ SACQ academic adjustment, social adjustment, and attachment than their international counterparts." Data from Tables 3 and 4 show this, and data from Table 5 show that those differences are statistically significant.

7.       Tables 7-11 are too many. Please, first of all, change the spacing between lines as they are too large. Include only the data where the differences/correlations are statistically significant. If you want to present the other information, you can include it as Supplementary Materials.

8.       In data analyses. I think the strategies for coping need to be discussed in relation to the student's nationality and the country they come from. Including all the students in a single discussion may not be relevant, since 91 of them are Asian. You could focus on the coping activities of Asian students from that university. At the very least, make a comparison between Asian students and others to see if the primary coping strategies are the same. If they are, you can discuss them together. This isn't the ideal approach, but it provides some rationale.

9.       Regardless, what you wrote in the Limitations section remains valid. Future discussions should include a cultural aspect of research. Additionally, it is important to examine whether the sociodemographic profiles of Asian versus other international students differ. There may be other variables that explain the results.

10.   The same consideration applies to the profile of American students. Differences between international students and Americans can be explained by some sociodemographic variables. Is the profile of American students similar to that of Asian students? If not, you could include these variables in your analysis to determine if they contribute to the observed differences in your scores and coping strategies.

I wish you success!

Reviewer 3 Report

Comments and Suggestions for Authors

This article explores a very relevant topic, namely mental health in international university students, focusing on the issue of changes due to the covid-19 pandemic. The theme of this article is interesting and timely, as students’ mental health is an increasing concern in our society. This research is formally correct and clear to the reader, although I suggest checking the manuscript for minor mistakes or inaccuracies (for example, LL 296 and 297 report an incorrect number of the number of subjects included; moreover, there is no correspondence between the caption and the heading of table 2). The text is complete and well structure, my only suggestion is to update the reference list (see for this purpose this recent paper on the topic, published in 2024 DOI: 10.3390/ijerph20054071). 
